# Spanning Fermi arcs in a two-dimensional magnet

Ying-Jiun Chen [1,2] ✉, Jan-Philipp Hanke[1,3], Markus Hoffmann [1,3], Gustav Bihlmayer [1,3], Yuriy Mokrousov [1,3,4], Stefan Blügel [1,3], Claus M. Schneider[1,2,5] & Christian Tusche [1,2] ✉

The discovery of topological states of matter has led to a revolution in materials research. When external or intrinsic parameters break symmetries, global properties of topological materials change drastically. A paramount example is the emergence of Weyl nodes under broken inversion symmetry. While a rich variety of non-trivial quantum phases could in principle also originate from broken time-reversal symmetry, realizing systems that combine magnetism with complex topological properties is remarkably elusive. Here, we demonstrate that giant open Fermi arcs are created at the surface of ultrathin hybrid magnets where the Fermi-surface topology is substantially modified by hybridization with a heavy-metal substrate. The interplay between magnetism and topology allows us to control the shape and the location of the Fermi arcs by tuning the magnetization direction. The hybridization points in the Fermi surface can be attributed to a non-trivial mixed topology and induce hot-spots in the Berry curvature, dominating spin and charge transport as well as magneto-electric coupling effects.

Classifying topological states of matter based on their global properties has matured into a pervasive scientific concept that provides an elegant interpretation of physical phenomena. Prominent examples include the recent discovery of axions in condensed matter[1,2], the quantum anomalous Hall effect[3,4], and the emergence of robust spin textures[5,6], all of which root in non-trivial characteristics of the wave function in magnetic solids. Ever since these striking advances, the intricate interplay between magnetism and topology is moving steadily into the focus of research, owing to bright promises for energy-efficient information processing[7,8] and brain-inspired computing[9]. To achieve these innovative functionalities, realizing non-trivial topological phases in two-dimensional (2D) magnets opens up a highly attractive route for, e.g. the low-dissipation control of magnetism mediated by mixed Weyl points[8]. Among such classes of intriguing 2D quantum materials are in particular layered van der Waals ferromagnets[10,11] and magnetic topological insulators[12,13], which received considerable attention lately.

Symmetries play a key role in identifying and understanding emergent phases of matter with complex topologies. For instance, time-reversal (TR) or inversion symmetries can manifest in protected exotic states in the electronic structure of topological insulators and Dirac semimetals[14–16]. If one of these protective symmetries is broken, the global properties of the material may change drastically as reflected by, for example, the splitting of individual Dirac points into pairs of Weyl points with opposite chirality. These points are linked by open Fermi arcs that emerge at the surface as a direct consequence of the non-trivial Fermi-surface topology[17–21]. Since ferromagnetism spontaneously breaks TR symmetry, controlling the magnetic order serves as an ideal means to create on-demand topological phase transitions and significantly alter the topology of the electronic states. Consequently, the concept of magnetic topological semimetals has recently attracted great research interest[13,22]. While the systems studied so far still belong to the class of three-dimensional (3D) Weyl semimetals with topological points located in 3D momentum space, the concept needs to be

[1]Peter Grünberg Institut, Forschungszentrum Jülich, 52425 Jülich, Germany. [2]Fakultät für Physik, Universität Duisburg-Essen, 47057 Duisburg, Germany. [3]Institute for Advanced Simulation, Forschungszentrum Jülich and JARA, 52425 Jülich, Germany. [4]Institute of Physics, Johannes Gutenberg University Mainz, 55099 Mainz, Germany. [5]Department of Physics, University of California Davis, Davis, CA 95616, USA. ✉e-mail: yi.chen@fz-juelich.de; c.tusche@fz-juelich.de

extended for low-dimensional systems. Despite the increasing importance of 2D magnetic materials[10,11,13,23–25], very little is known about the interplay between magnetism and topology in such low-dimensional magnets.

Here we focus on a 2D topological ferromagnet that consists of two monolayers (MLs) Fe with in-plane magnetization, grown epitaxially on a W(110) substrate. It was recently predicted that bcc Fe is a topological metal as its electronic structure hosts emergent Weyl points and arc-like resonances on the (110) surface connecting them[26,27]. However, these non-trivial features usually do not affect macroscopic phenomena as they are located far away from the Fermi energy. By interfacing a thin Fe film with a heavy metal providing strong spin-orbit coupling (SOC), we promote this intrinsic hidden topology and bring the associated topological features to the surface realm, with immediate consequences for macroscopic observables.

Our results show that giant open Fermi arcs are created at the surface of ultrathin hybrid magnets. The Fermi-surface topology of an atomically thin ferromagnet is substantially modified by the hybridization with a heavy-metal substrate, giving rise to Fermi-surface discontinuities that are bridged by the Fermi arcs. Due to the interplay between magnetism and topology, we can control both the shape and the location of the Fermi arcs by tuning the magnetization direction. The hybridization points in the Fermi surface can be attributed to a non-trivial mixed topology and induce hot-spots in the Berry curvature, dominating spin and charge transport as well as magneto-electric coupling effects.

## Results and discussion

### Emergence of Fermi arcs in a magnetic 2ML Fe film

Owing to the different global properties of the topological metal Fe and the heavy-metal substrate W, the composite system is ideally suited to experimentally realize an interface-driven topological quantum phase in the presence of broken TR symmetry and strong SOC. We unveil the non-trivial topological state realized in this 2D magnet by studying the corresponding Fermi surface, using a recently available technique known as spin-resolved momentum microscopy[28,29]. Figure 1a presents the principle layout of the experimental geometry for our spin- and orbital-resolved photoemission study. The momentum

microscope simultaneously collects photoelectrons over the full solid angle above the sample, such that spin- and momentum-resolved photoemission experiments over the whole Brillouin zone can be performed (see Methods)[28,29]. This technique gives comprehensive and intuitive access to the electronic structure of materials and has advanced the frontiers in visualizing the orbital fingerprint throughout the whole Fermi surface by circular and linear dichroism[29,30]. Combined with the recent groundbreaking invention of an imaging spin detector[31,32], this approach offers full access to detailed spin-resolved Fermi surfaces that were previously inaccessible by conventional photoelectron spectroscopy[28,33].

Figure 1c shows the spin-resolved momentum map that we obtain at the Fermi energy, revealing the appearance of prominent surface states in the highlighted regions on both sides of the momentum map. These open arcs with a crescent-moon shape exhibit a high spin polarization. By comparing our experimental results to first-principles calculations (see Fig. 3a and Supplementary Fig. 1), we identify these arcs as true surface states that exist as open segments only in a limited region of the surface Brillouin zone, and which terminate at the intersections with the interior states. Interestingly, the location of the surface Fermi arcs on 2 MLs Fe/W(110) corresponds to the region where two non-trivial Fermi-surface electron pockets with opposite Chern number emerge in bulk bcc Fe. In our experiments we observe these topological electron pockets in bulk Fe (see Supplementary Fig. 2 and Supplementary Note 2), in good agreement with previous reports[26,27]. The relatively weak SOC in bulk Fe and the overlap with bulk states gives rise to only unstable Fermi-arc surface resonances, which thus have not be observed in an experiment so far[27]. Due to the increased SOC mediated by the heavy-metal substrate tungsten, one can anticipate that these chiral band degeneracies of iron might promote the observed surface arcs as outlined below. The observation of emerging surface arc states in a 2D ferromagnet provides a rich platform for exploring topological magnets beyond conventional Weyl semimetals. As the observed open surface states share strong analogues with their cousins in 3D topological materials[17–21], we refer to them as Fermi arcs in a 2D topological ferromagnet.

To understand the origin of these non-trivial states, we repeat the measurements for a reference system where the thin ferromagnet is

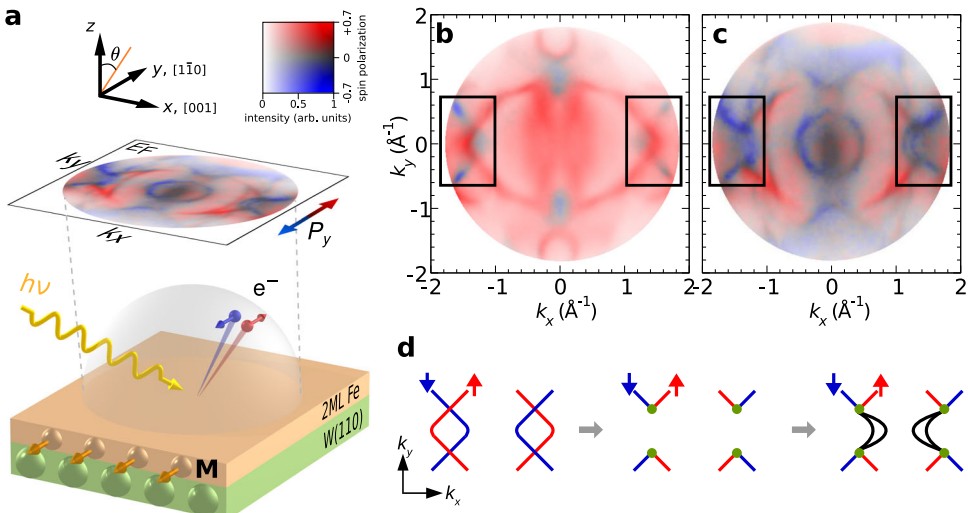

**Fig. 1 | Fermi arcs in a 2D topological ferromagnet. a** Experimental geometry for the spin-, orbital-, and momentum-resolved photoelectron study of the Fermi surface. The magnetization **M** and the spin polarization $P_y$ are oriented in-plane along the $y$ axis, i.e., the crystallographic [1$\bar{1}$0] direction. **b, c** Spin-resolved photoemission momentum maps at the Fermi energy $E_F$ for **b** 12 MLs and **c** 2 MLs Fe films grown on W(110), measured with a photon energy of $h\nu = 50$ eV for a sample

magnetization pointing into $-y$ direction. The spin polarization $P_y$ is indicated by red and blue colours, and the colour strength encodes the intensity. **d** Schematic evolution of the Fermi-surface topology for the appearance of surface arcs upon breaking TR symmetry. Red (blue) colour denotes spin-up (spin down) electron states, the green circle denotes the spin-orbit mediated hybridization, and black lines indicate the emergent surface Fermi arcs.

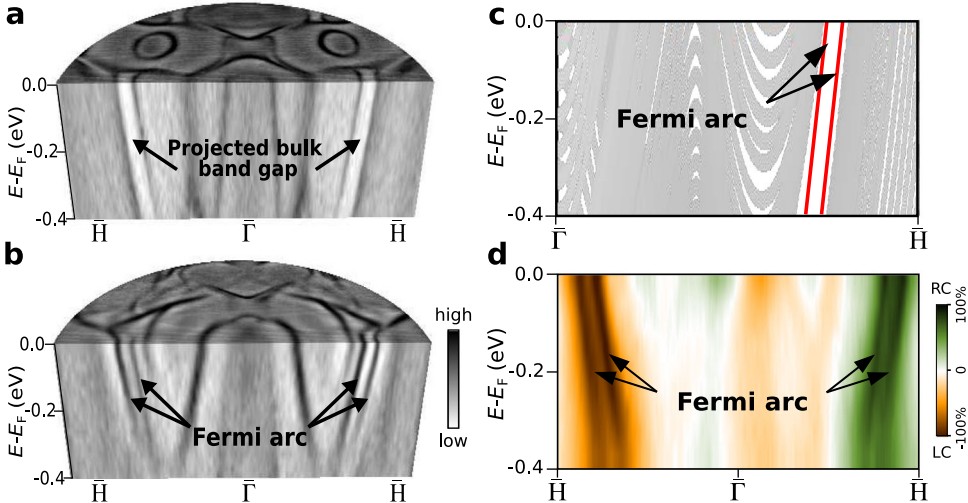

**Fig. 2 | Fully helicity-selective surface arcs.** Band dispersions measured on **a** W(110) and **b** 2 MLs Fe/W(110) along $\overline{H} - \overline{\Gamma} - \overline{H}$ at the photon energy $h\nu = 50$ eV for the magnetization direction $\mathbf{M}_{+y}$. **c** Schematic diagram of the two co-propagating surface Fermi arcs (red lines) in the 2D topological ferromagnet, which appear inside the spin-orbit-induced projected band gap (large white area) of the substrate. **d** The measured circular dichroism reveals that the surface arcs in 2 MLs Fe/W(110) are 100% helicity-selective, with opposite helicity between the two sides of the momentum map. The labels RC and LC denote right and left circular polarization of the light, respectively, and the colour code represents the magnitude of the circular dichroism signal.

replaced with a thick Fe film of 12 MLs. In that case, the surface Fermi arcs disappear and two states with opposite spin polarization emerge instead, crossing each other without any hybridization (see Fig. 1b). This bulk-like behaviour is drastically different from the electronic structure of the atomically thin magnetic films that are strongly affected by the sizeable SOC of the substrate.

Anc important condition for the formation of the Fermi-arc surface states is the topological transition of the Fermi-surface that is triggered by the strong SOC brought about by the heavy-metal substrate. The changed topology of the Fe Fermi-surface sheets can be understood as follows: due to the presence of magnetic exchange interaction, the spin degeneracy is lifted in ferromagnetic Fe. In the limit of sufficiently large exchange splitting and negligible SOC, majority and minority spin states intersect each other as shown in Fig. 1b. Thus, without SOC, the majority (↑) and minority (↓) bands are decoupled and can not interact.

Figure 1d illustrates how SOC leads to the emergence of the Fermi arcs in the iron layers. Without sizeable SOC, majority and minority Fermi-surface contours can intersect without hybridization (left panel). The hybridization of Fe wave functions with the heavy-metal substrate tungsten leads to a strong SOC contribution, such that majority and minority spin states can mix at the crossing points (green circles in Fig. 1d). As a result, the degeneracy at the intersection region is removed and a spin-orbit induced local gap leads to a discontinuity of the formerly closed contours of the majority and minority Fermi-surface sheets (centre panel). This SOC gap opening changes the topology of the Fe Fermi surface, contravening that the iso-energy contour at the Fermi level of any conventional metal needs to be closed by a well-defined quasiparticle dispersion. The gap thus needs to be bridged by the emergent Fermi arcs (black lines) as illustrated in the right panel of Fig. 1d.

Unlike closed loop surface states of ordinary metals, where quasiparticles travel merely on the surface, Haldane[18] has shown that the Fermi arc surface states act as surface conduits that adiabatically transfer quasiparticles between topologically non-trivial sheets of the metallic Fermi surface, which are otherwise disjoint. Thus, the open ends of the Fermi arcs are connected to these different Fermi-surface sheets such that the incomplete surface band ends at the respective intersection points. This mechanism allows the system to maintain a common chemical potential across the apparently disconnected

Fermi-surface sheets, and can provide an experimental diagnostic evidence for the existence of a Weyl metal[18,19]. In 3D materials, this is further known to result in highly non-trivial consequences on the spin- and orbital degrees of freedom in the system[18,19]. As we demonstrate below, similar conclusions likewise apply in the present case of a topological 2D ferromagnet.

As it is important to know whether the surface arcs lie in a band gap of the substrate, we compare the results of surface band structure measurements without and with the thin magnetic layer on top of the substrate. As shown in Fig. 2a, an energy gap is found in W(110) along the $\overline{H} - \overline{\Gamma} - \overline{H}$ line, which has been ascribed to a partial bulk band gap that is induced by strong SOC, thereby resembling the relativistic fundamental gap in topological insulators[34–38]. When an atomically thin Fe film is grown on the W(110) substrate, the surface arcs with linear dispersion co-propagate inside this projected bulk band gap (see Fig. 2b, c). This result exhibits an analogy to the quantum anomalous Hall (QAH) effect in doped topological insulators where chiral edge states co-propagate in the same direction[39]. In that case, an opposite handedness is observed at opposite sides of the Brillouin zone with reversed velocity. Such a mechanism is general for realizing QAH phases with combined spontaneous magnetic moments and SOC in the absence of an external magnetic field[3,4].

**Magnetic tuning of Fermi-arc states**

Figure 3a shows the calculated Fermi surface of 2ML Fe on W(110). The colour in Fig. 3a represents the degree of states' localization in the Fe layers. These dark coloured states are essentially derived from Fe, but can couple to the states of the substrate. Thus, the wave functions of 2ML Fe hybridize and extend into the continuum of the Bloch states of the W substrate. This results in the Fermi-surface contour being considerably different from the one of the corresponding freestanding Fe film, shown in Supplementary Fig. 3a. The calculation further reveals that the emergent Fermi arcs in the highlighted regions are located within the projected band gap of the bulk W(110) substrate. The Fermi arcs therefore are characterized as true surface states, which have no bulk correspondence, in contrast to the other interior states in the 2ML film. As clearly evident, the substrate plays a crucial role in promoting the non-trivial features of the hidden topology in Fe. In particular, no Fermi arcs emerge for a freestanding 2ML Fe(110) film (see Supplementary Fig. 3a). This result is in line with previous reports of

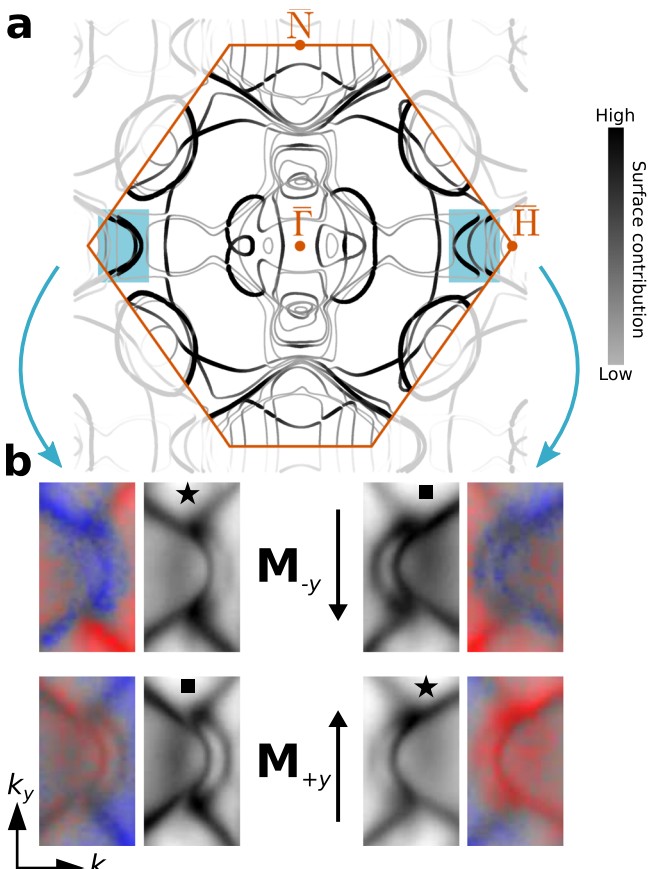

**Fig. 3 | Fermi-arc switching induced by magnetization reversal. a** Theoretical Fermi surface of 2 MLs Fe/W(110) for the in-plane magnetization $\mathbf{M}_{-y}$. Colours represent the degree of the states' localization in the magnetic layers, and the shaded regions highlight the emergent Fermi arcs. **b** Measured spin-resolved and spin-integrated surface arc states in the highlighted regions of **a** for the two magnetization directions $\mathbf{M}_{+y}$ and $\mathbf{M}_{-y}$ that are parallel and anti-parallel to the $y$ direction, respectively. The shape and the spin polarization of the strongly asymmetric Fermi arcs on opposite sides of the momentum map is interchanged if the magnetization direction is reversed as visualized by the black star and square symbols. The spin polarization $P_y$ is indicated by red and blue colours, and the colour strength encodes the intensity.

freestanding single monoloayer Fe films where no Fermi arcs were observed, likewise[40,41].

Quantum phenomena that result from the breaking of TR symmetry offer an innovative platform for on-demand design and non-volatile switching of physical properties by controlling the direction of the sample magnetization[8,42,43]. To explore the magnetization response of the observed arc states, we performed momentum microscopy experiments for both magnetization orientations, parallel and anti-parallel to the $y$ direction as denoted by $\mathbf{M}_{+y}$ and $\mathbf{M}_{-y}$, respectively. As apparent from Fig. 3, these orientations of the magnetization break the two-fold symmetry of the (110) surface, leading to a Fermi surface that exhibits mirror symmetry with respect to the $k_y = 0$ mirror plane, whereas non-equivalent features occur with respect to the $k_x = 0$ plane. Remarkably, among all states at the Fermi surface, the open arcs show the most striking asymmetry of the photoemission intensity and the shape between the left and right side of the momentum map. Such magnetization-dependent asymmetries require the presence of SOC[44]. Here, the strong SOC is mediated through hybridization with substrate states in the regions of the Fermi arcs. The pronounced $\mathbf{k}$-dependent relativistic splitting is then promoted by both, the broken TR symmetry and missing spatial inversion symmetry at the interface. This mechanism is reminiscent of the physical process that gives rise to the

interfacial Dzyaloshinskii-Moriya interaction, leading to chiral and non-collinear magnetic structures. By virtue of the spin-resolved data, Fig. 3b, a closer look at the surface arcs reveals that the sign of their spin polarization $P_y$ changes upon magnetization reversal as the surface arc topology is interchanged between the two sides of the momentum map.

## Spin texture and Berry curvature

In 3D metallic ferromagnets, most of the states are nearly pure spin states, namely, either spin-up (majority) or spin-down (minority) states with respect to the sample magnetization. Due to the broken inversion symmetry at interfaces with heavy-metal substrates that provide SOC, non-collinear spin textures may be favoured over collinear configurations. To determine the full spin texture of the surface arcs, we have measured in our momentum microscopy experiments also the in-plane spin polarization $P_x$ along the $x$ direction, which is orthogonal to the sample magnetization $\mathbf{M}_{-y}$ of the 2D topological ferromagnet. According to our first-principles calculations, the states carry no spin polarization perpendicular to the film plane. As shown in Fig. 4a, the Fermi arcs exhibit a surprisingly sizable spin polarization $P_x$ that amounts to as much as 50% of the signal $P_y$. A prominent variation of spin polarization is observed in the right pair of arcs, indicating a significant non-collinearity of the spin texture. The variation of the $P_x$ and $P_y$ spin components in the upper and lower part of the arc can be seen quantitatively in horizontal spin-resolved intensity profiles across the arc (see Supplementary Fig. 6 and Supplementary Note 6). In particular the $P_x$ component undergoes a change of sign along the arc, in good agreement with the theoretical spin texture in Fig. 4b. By contrast, this non-collinearity is less pronounced for the left pair of arcs. In good qualitative agreement with our experimental findings, the theory confirms for the right arcs a pronounced deviation of the local spin orientation from the vertical magnetization direction.

In contrast to the Fermi arcs that appear in Weyl semimetals with TR symmetry[17,19], our results for 2 MLs Fe/W(110) demonstrate that not only the shape but also the spin texture of the Fermi arcs is asymmetric with respect to the centre of the surface Brillouin zone due to the global breaking of TR symmetry. Clearly, we find that the full spin texture uncovers a prominent degree of non-collinearity in the Fermi arcs, distinguishing them from the regular ferromagnetic states with collinear spin polarization. Furthermore, in contrast to topological surface states on many topological insulators and related layered materials, where the spin texture is locked to different p-orbital symmetries[45,46], the Fermi-arc states of 2ML Fe/W(110) are composed of d-orbitals. In particular, our first-principles calculations show that one branch of an arc has a major $d_{xy}$ orbital character, and another is of $d_{yz}$ orbital symmetry. This sets the Fermi-arc states of Fe/W apart from conventional topological materials. In particular the complex spin texture along the arcs is already present in the ground state, as one can also see from the calculation in Fig. 4b.

Since the evanescent wave functions of the surface arcs inherit their velocity from the interior states of the film, they are attached tangentially as they approach the termination points, as observed in Fig. 4b. Analogously, the complex interplay between topology and magnetism is well known to locally imprint also the orbital properties of individual states close to topological phase transitions[13]. Indeed, our first-principles calculations in Supplementary Fig. 4 demonstrate that the distinct topology of the observed arc-like surface states is not only correlated with the spin polarization but manifests as well in unique changes of the orbital character, as the corresponding states evolve into the termination points. Consequently, our experimental and theoretical results unveil that the rapid variation in the spin and orbital texture reflects the abrupt transition from the genuine surface arcs into the interior states at the intersection points, in close resemblance to the orbital fingerprint of topological Fermi arcs in a Weyl semimetal[19].

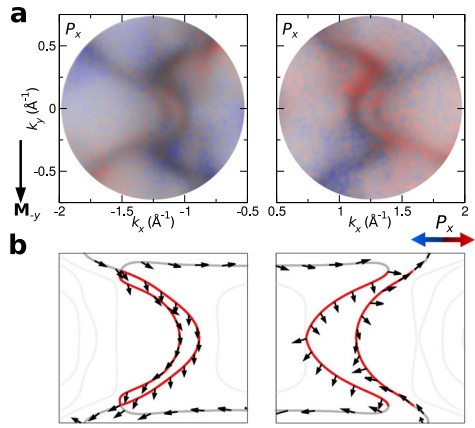

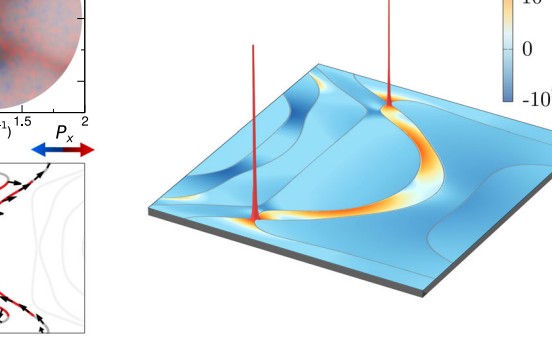

**Fig. 4 | Spin texture and Berry curvature of the Fermi arcs in the 2D topological ferromagnet. a** Measured spin-resolved Fermi arcs in 2 MLs Fe/W(110) on the left and on the right side of the momentum map. Colours indicate the in-plane spin component along the $x$ direction, which is orthogonal to the sample magnetization $\mathbf{M}_{-y}$. The spin polarization $P_x$ is indicated by red and blue colours, and the colour strength encodes the intensity. **b** Arrows denote the complete theoretical spin texture in momentum space, revealing a prominent non-collinearity for the Fermi arcs (red) as compared to the interior states (grey). **c** Distribution of the theoretical momentum-space Berry curvature $\Omega_{xy}^{\mathbf{kk}}$ of all occupied bands in 2 MLs Fe/W(110), around one of the pairs of Fermi arcs. A pseudo-logarithmic colour scale and a linear height field are used to represent the local geometrical curvature, and the Fermi surface is indicated by grey lines.

The Fermi-arc states in usual Weyl semimetals inherit chiral properties from the topological nodal points that they connect. To explore whether the observed Fermi arcs in the present system with broken TR symmetry reflect this chiral nature as well, we carried out momentum-resolved photoemission experiments using left- and right-circularly polarized light, without changing the overall experimental geometry. Figure 2d presents our results for the circular dichroism (CD), which uncovers that the difference in excitation for light with different sense of circular polarization amounts to as much as 100% for the arc states. As a consequence, photons with different helicity excite selectively the Fermi arcs either at the right side or at the left side of the momentum map. The sign and magnitude of CD remains unchanged when the spin texture is reversed. This implies that the dichroic signal mainly originates from the orbital part of the wave functions, being independent of the initial-state spin polarization. The two co-propagating surface arcs are the sharpest observed features in the CD signal in Fig. 2d, in contrast to the other states with relatively broad CD spectral features and substantially lower CD magnitude. Since the final states, that in principle might contribute to the CD signal, are significantly broadened at energies 50 eV above the Fermi level such sharp spectral features with a 100% CD in the entire binding energy range can be mainly attributed to the Fermi arcs.

In typical Weyl semimetals, the CD signal correlates with the chirality of nodal points, reflecting non-trivial global properties of the Bloch states in momentum space as encoded locally in the Berry curvature[47,48]. The latter geometrical quantity is a key ingredient in the interpretation of the intrinsic contribution to the anomalous Hall effect. We point out that the regions with the largest CD signal in Fig. 2d, correspond exactly to the emergent Fermi arcs, and provide substantial contributions to the Berry curvature in momentum space, as we will see below. In particular, it recently has been shown that the CD of Fermi arc states of conventional Weyl semimetals provides a direct fingerprint of the orbital angular momentum and the respective intrinsic chirality of the Weyl states[49]. Transferring this concept to the considered case of the 2D topological ferromagnet, our results suggest that a similar fingerprint in the CD signal likewise applies, here.

In order to verify the microscopic insights revealed by our dichroism measurements, we performed theoretical first-principles calculations of the momentum-space Berry curvature for the considered 2D topological ferromagnet. Figure 4c presents the resulting distribution of the Berry curvature summed over all occupied bands, which underlies many macroscopic properties such as orbital magnetism and intrinsic Hall responses. While there are several minor background contributions, the most prominent features stem from states that form the observed Fermi arcs, with strong hot-spots where the surface arcs connect to the attachment points, see also Supplementary Fig. 5.

## Mixed Weyl points in a composite phase space

Conventional 3D Weyl materials are well described in the space of the 3D crystal momentum $(k_x, k_y, k_z)$, where the Weyl nodes are located. In a 2D ferromagnet, as in the present case of 2ML Fe/W(110), this description breaks down since the momentum coordinate $k_z$ is not a good quantum number in the low-dimensional system. Beyond this conventional Weyl picture, the emergence of topological points can be understood when we consider the additional degree of freedom of the magnetization, that is already the source of broken TR symmetry. In this description, the regular $(k_x, k_y, k_z)$ momentum space is replaced by a mixed phase space $(k_x, k_y, \hat{\mathbf{m}})$, where $\hat{\mathbf{m}}$ denotes the magnetization direction[8,13]. The observed pronounced change of Fermi-surface states induced by changes of the magnetization, thus, can be interpreted as a direct signature of the topology of this mixed phase space. To investigate the origin of the open Fermi arcs at the surface of the 2D magnetic layer, we include the magnetization direction and the mixed components of the Berry curvature into the topological analysis.

The magnetization direction $\hat{\mathbf{m}} = (\sin \theta, 0, \cos \theta)$ of a 2D magnet encloses the angle $\theta$ with the $z$-axis. When inversion symmetry would be preserved, Weyl points in this mixed space would extend along the $\theta$ coordinate axis[13]. Since in the Fe film on the W(110) substrate simultaneously inversion symmetry as well as time-reversal symmetry is broken, a pair of the Weyl nodes can distribute in the $(k_x, k_y)$ plane. This can be observed in Fig. 5a, where the sign of the mixed Berry curvature $(\Omega_{yx}^{\widehat{\mathbf{mk}}})$ changes sharply across the intersection points with the Fermi arcs in the $k_x$–$k_y$ plane, while the momentum Berry curvature $(\Omega_{xy}^{\mathbf{kk}})$ changes the sign with the magnetization direction at $\theta \approx 88°$. The Berry curvature changes sign when the topological phase transition occurs. Figure 5b reveals that the Berry curvature field $\Omega$ acquires a monopole-like distribution around these mixed Weyl points, located at $\theta \approx 88°$. This can be directly related to the presence of topological charges in the composite phase space, which act as sources and sinks of the curvature field, and drive the emergence of the Fermi-arc states. For conventional 3D Weyl semimetals it is generally believed that the

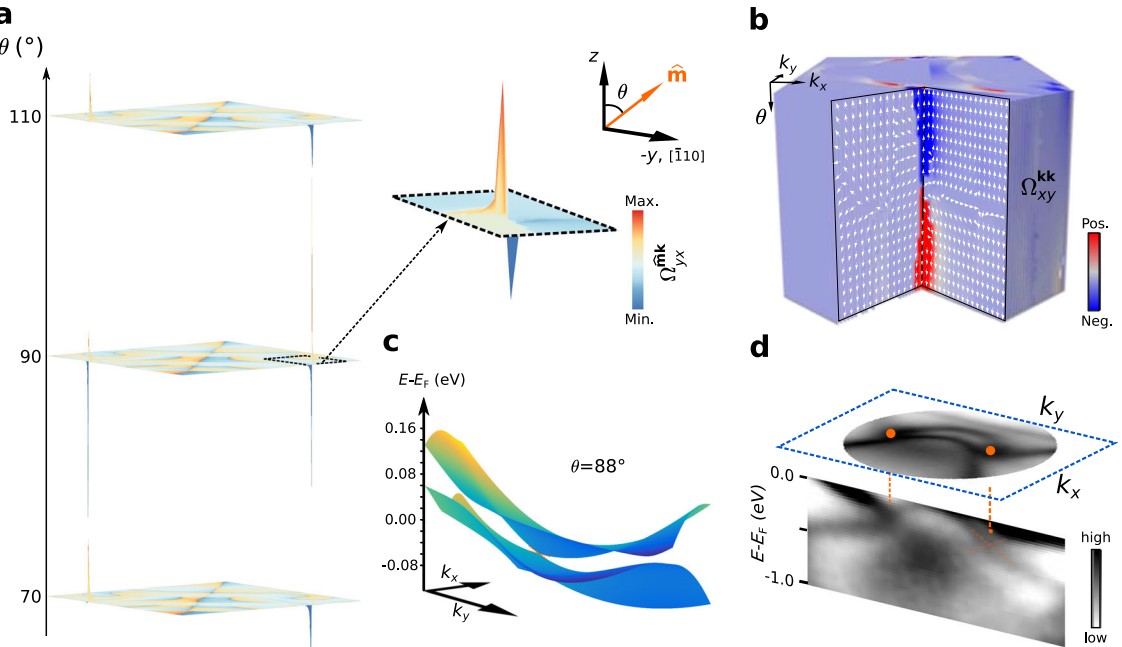

**Fig. 5 | Mixed Weyl cones in a 2D topological ferromagnet. a** The mixed Berry curvature distribution $\Omega_{yx}^{\hat{m}k}$ as a function of the magnetization direction ($\theta$); inset shows a magnified view near one end of the surface arc to highlight the sharp change in its sign across the projected mixed Weyl nodes. A logarithmic colour scale and a linear height field are used, where red (blue) denotes positive (negative) values. **b** Monopole-like field of the phase-space distribution of the Berry curvature $\Omega = (-\Omega_{yy}^{\hat{m}k}, \Omega_{yx}^{\hat{m}k}, \Omega_{xy}^{kk})$ near the end points of the Fermi arcs at $\theta = 88°$. Arrows indicate the direction of the Berry curvature field, and the colour scale from blue

(negative, Neg.) to red (positive, Pos.) encodes the momentum Berry curvature $\Omega_{xy}^{kk}$ in the composite space of **k** and $\theta$. **c** Calculated band dispersion of the mixed Weyl cone emerging at $\theta = 88°$. Colour shading indicates the binding energy ranging from −0.08 eV to 0.16 eV. **d** Measured band dispersions along a straight cut through two intersection points indicated by orange dots. Surface Fermi arcs spanning in the $k_x$–$k_y$ plane connect the projection of the mixed Weyl nodes. The experimental observation of band crossings near the Fermi level agrees well with theoretical predictions in **c**. The orange dashed lines are a guide to the eye.

number of Fermi arcs corresponds to the chiral charge $\mathcal{C}$ of the Weyl point to which the arcs are attached. For instance, TaAs was found to host Weyl points with chiral charge $\mathcal{C} = \pm 2$ attached to two Fermi arcs[50], and the type-II Weyl semimetals $Mo_xW_{1-x}Te_2$[51] and $1T_d$-MoTe$_2$[52] hosts one Fermi arc and Weyl points with a charge of $\mathcal{C} = \pm 1$. Applying this concept to the present case of the mixed Weyl points we thus find a charge $\mathcal{C} = \pm 2$.

In order to identify the source of these topological charges we followed the evolution of the electronic structure of the 2ML Fe film with the magnetization direction ($\theta$). As shown in Fig. 5c, strongly tilted cones emerge at band touching points of electron and hole pockets at $\theta \approx 88°$. These degeneracy points, located at the Fermi energy, form the mixed Weyl points that give rise the charges that we have observed in the mixed phase space in Fig. 5a and b. In good agreement with the calculation, the appearance of the degeneracy points is confirmed by our experimental data. Figure 5d shows the measured spectral function, where the degeneracy points appear at a slightly lower binding energy of 200 meV below the Fermi level. This location is still close enough to the Fermi level that the peaks in the Berry curvature that are associated with the mixed Weyl nodes are expected to significantly contribute to magneto-transport properties, like the anomalous Hall conductivity of the film[8]. As clearly visible in our calculations (Fig. 5c) and the experimental data (Fig. 5d), the open-loop Fermi arcs are connected to the projections of these mixed Weyl nodes at $\theta \approx 88°$ around the Fermi energy.

The presence of such emergent monopoles in the mixed phase space close to the Fermi energy has been previously found as the source of a particularly strong Dzyaloshinskii-Moriya interaction (DMI)[8,13]. Our present results therefore provide an explanation for prior reports of an exceptionally large DMI in 2ML Fe/W(110)[53,54], which thus can be directly related to the presence of the non-trivial Fermi-surface topology in this system. Correspondingly, the exotic complex

topology in the high-dimensional space boosts the variation of the spin and orbital character of Fermi arcs.

As a consequence, these regions of large geometrical curvature constitute the most important sources of macroscopic effects related to charge transport and orbital properties, rendering the emergent Fermi arcs the key ingredients in understanding these phenomena in 2D topological ferromagnets. Moreover, owing to the distinct nature of the open arcs as compared to the trivial Fe and W(110) states, we expect that the former activate additional current-induced field-like torques that act on the magnetization.

In summary, we have identified the appearance of open surface states in a 2D topological ferromagnet, resulting from the complex interplay between an itinerant ferromagnet and a heavy metal with strong spin-orbit coupling. Our results suggest that the occurrence of non-trivial Fermi-surface topologies is much more abundant in conventional magnetic material than commonly expected. An advanced materials design that tailors the concurrence of symmetry, exchange, and SOC can bring such hidden topologies to the surface, with direct implications on exotic magneto-transport properties. Uncovering these principles can promote innovative technologies based on topological spin-orbitronics, where non-trivial topological states are controlled on demand through the magnetization direction.

## Methods
### Spin-resolved momentum microscopy
Spin- and momentum-resolved photoelectron spectroscopy experiments were carried out at the NanoESCA beamline[55] of the Elettra synchrotron in Trieste (Italy), using p- and s-polarized photons at the photon energy $h\nu = 50$ eV. All measurements were performed while keeping the sample at a temperature of 130 K. Photoelectrons emitted into the complete solid angle above the sample surface were collected using a momentum microscope[28,29]. The momentum microscope

directly forms an image of the distribution of photoelectrons as function of the lateral crystal momentum ($k_x,k_y$) that is recorded on an imaging detector[28,29].

An imaging spin filter based on the spin-dependent reflection of low-energy electrons at a W(100) single crystal[31] allows the simultaneous measurement of the spin polarization of photoelectrons in the whole surface BZ. Images were recorded after reflection at a scattering energy at 26.5 and 30.5 eV that changes the spin sensitivity $S$ of the detector between 42% and 5%, respectively. From these images, the spin polarization at every ($k_x,k_y$) point in a momentum map at a certain binding energy (e.g. Figs. 1 and 3–4 in the main manuscript) is derived as $P(k_x,k_y) = [\mathcal{I}_{26.5\,\mathrm{eV}}(k_x,k_y) - \mathcal{I}_{30.5\,\mathrm{eV}}(k_x,k_y)]/[S_{26.5\,\mathrm{eV}} \cdot \mathcal{I}_{30.5\,\mathrm{eV}}(k_x,k_y) - S_{30.5\,\mathrm{eV}} \cdot \mathcal{I}_{26.5\,\mathrm{eV}}(k_x,k_y)]$, where $\mathcal{I}_{\mathrm{ES}}$ denotes the measured intensity image at scattering energy ES normalized by the respective reflectivity as measured from the clean Cu(100) surface[31–33].

The W(100) crystal of the spin filter was prepared by following the same procedure as for W(110) below. This standard procedure is known to lead to clean tungsten surfaces[31,32]. At a pressure of $1 \times 10^{-10}$ mbar inside the spin-filter chamber during the measurements, the analyser-crystal could be used for 2 h. This time frame allowed for collecting 2D spin-resolved momentum maps at several energies[33].

In all measurements the energy resolution of the momentum microscope was set to 50 meV. The momentum resolution for spin-integrated measurements was $\Delta k_\| = 0.02\,\text{Å}^{-1}$. For spin-resolved measurements, the momentum resolution is determined by properties of the W(100) crystal and depends on the field-of-view of the momentum image[31]. For data displayed in Figs. 1b, c and Fig. 3, the momentum resolution was $\Delta k_\| = 0.05\,\text{Å}^{-1}$. Due to a smaller 2D field of view, for Fig. 4a the momentum resolution was $\Delta k_\| = 0.02\,\text{Å}^{-1}$.

### Growth of the quasi-2D Fe monolayer on W(110)

For the growth of 2 monolayers (MLs) Fe on W(110), the primary important step is to prepare a clean tungsten surface. The procedure for cleaning W(110) consists of two steps: (i) cycles of low temperature flash-heating (T ∼ 1200 K) at an oxygen partial pressure of $P_{O_2} = 5 \times 10^{-8}$ mbar to remove the carbon from the surface, and (ii) a single high temperature flash (T ∼ 2400 K) to remove the oxide layer[32]. The cleanliness of the W(110) surface was checked by low-energy electron diffraction (LEED) and Auger electron spectroscopy. Iron films with a thickness of 2 MLs were deposited in situ by molecular beam epitaxy from a high-purity Fe rod, heated by electron bombardment, onto the clean W(110) single crystal at a substrate temperature of 350 K.

Before carrying out spin-resolved momentum microscopy measurements, the samples were uniformly magnetized along the quantization axis of the spin filter, being the same direction as the easy magnetization direction ($\overline{\Gamma} - \overline{N}$, $k_y$) of 2 MLs Fe films on W(110). We see that most electronic states are either of majority (spin up) or minority (spin down) character with the spin polarization along the $k_y$ axis as shown in Figs. 1 and 3 of the main manuscript. In order to measure the transverse component of the spin polarization parallel to the $k_x$ ($\overline{\Gamma} - \overline{H}$) direction of the Fe films, we rotated the sample around 90°. The $k_x$ direction of the sample then is parallel to the quantization axis of the spin filter (see Fig. 4 of the main manuscript).

### First-principles calculations

Using the full-potential linearized augmented-plane-wave (FLAPW) method as implemented in the FLEUR code (See https://www.flapw.de), we performed density functional theory calculations of 2 MLs Fe on 7 layers of W(110). We adopted the structural parameters given in ref. 56. Exchange and correlation effects were treated within the local-density approximation (LDA)[57]. The spin-orbit interaction was included self-consistently and the magnetization direction was chosen according to the experimentally observed easy axis [1$\bar{1}$0]. The plane-wave cutoff was chosen as $k_{\max} = 4.2\,a_0^{-1}$ with $a_0$ as Bohr's radius, and the muffin-tin radii for Fe and W were set to 2.1 and 2.5 $a_0$, respectively.

To determine the Fermi surface with high precision, we calculated the eigenspectrum on a dense uniform mesh of 256 × 256 **k**-points in the full Brillouin zone. The Fermi level crossings were then obtained by triangulation. To improve the agreement with experiment, we shifted the position of the Fermi level in our calculations downwards by 70 meV as compared to the theoretical value for the Fermi energy.

Based on the converged electronic structure, we used the wave-function information on a uniform mesh of 8 × 8 **k**-points in order to construct a single set of 162 maximally localized Wannier functions (MLWFs)[58,59] out of 227 Bloch states via the WANNIER90 program[60,61]. In this procedure, we projected initially onto $d$-orbitals and $sp^3d^2$-hybrids, and we chose a frozen window that extends up to 1.5 eV above the Fermi level. The resulting MLWFs allow us to efficiently sample the full Brillouin zone via an accurate Wannier interpolation[62] that grants access to spin- and orbital-resolved properties, as well as to the Berry curvature.

## Data availability

The data associated with the figures of the main text have been deposited in the Jülich DATA repository under the address (https://doi.org/10.26165/JUELICH-DATA/CXWKMJ). The data that support the findings of this study are available from the corresponding author upon reasonable request.

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

## Acknowledgements

Y.-J.C and C.T. thank the staff of Elettra for their help and hospitality during their visit in Trieste, and beamline staff M. Jugovac, G. Zamborlini, V. Feyer (PGI-6, FZ-Jülich), and T. O. Menteş (Elettra) for their assistance during the experiment and providing the W(110) crystal. Y.-J.C., C.T., and C.S. gratefully acknowledge funding by the BMBF (Grant No. 05K19PGA). S.B. and Y.M. acknowledge support by the Deutsche

Forschungsgemeinschaft (DFG) through the Collaborative Research Centre SFB 1238 (Project C1) and Priority Programm SPP 2137. M.H., Y.M., and S.B. acknowledge funding from the DARPA TEE program through grant MIPR (#HR0011831554) from DOI. Y.M. acknowledges the Deutsche Forschungsgemeinschaft (DFG,German Research Foundation) - TRR 288 - 422213477 (project B06)and DFG Project 1731/10-1 for funding. We also gratefully acknowledge the Jülich Supercomputing Centre and RWTH Aachen University for providing computational resources under projects jara0197, jias1f, and jiff40.

## Author contributions

Y.-J.C. and C.T. carried out the experiment and analyzed the experimental data. J.-P.H. and M.H. carried out first-principle calculations and analyzed the theoretical data with assistance by G.B. and Y.M. Y.-J.C. drafted the manuscript with assistance by C.T. and J.-P.H. C.T. designed and coordinated the research together with C.M.S. and S.B. All authors discussed the results and contributed to the manuscript.

## Funding

## Competing interests

The authors declare no competing interests.
