## [Peer Review File · Nature Communications]

Reviewers' Comments:

Reviewer #1:

Remarks to the Author:

This manuscript presents the spin-polarized Fermi arc emerging in 2MLs Fe films on W(110) due to a magnetic Weyl state. This feature has been successfully demonstrated by ARPES with a momentum microscope method equipped with a 2D spin detector. Weyl states are first demonstrated by ARPES for TaAs with the inversion symmetry breaking. Currently, magnetic Weyl states are being intensively studied in the field of condensed matter physics. In the states, the time-reversal symmetry breaking is the trigger for the realization of Weyl states. The magnetic Weyl states have functionalities being controllable by external magnetic field thus more advantageous for applications over those of inversion symmetry breaking systems. Such magnetic Weyl states have been experimentally demonstrated for 3D crystals so far. The current work is unique in two ways: one is that Weyl states are generated in the form of films with two-dimensionality. The second is that the sample is made of conventional magnetic elements Fe even though it is assisted by W substrate with strong spin-orbit interaction. The data has very high quality, and the Fermi arc featuring Weyl states is convincingly revealed. In particular, the switching of spin polarization by flipping the magnetization direction is an excellent demonstration of engineering topological state with conventional magnetic metals which was achieved owing to the state-of-arts ARPES technique.

I will request the following issues to be addressed before considering publication of the manuscript:

- 1) In the 2 MLs Fe films on W(110), not only time-reversal symmetry but inversion symmetry are broken. From the current manuscript, I could not figure out the influence of each symmetry breaking on the spin-polarized Fermi arc. The authors should discuss this in detail.
- 2) Throughout the entire manuscript, the arguments are given only with the presentation of photoelectron intensity maps. I request that the authors also plot up and down spin-resolved spectra along momentum cuts across the Fermi arc and discuss the details on the ky component in Fig. 3 and kx component in Fig.4.
- 3) I also wonder if the authors also measured the spin-polarized spectra as a function of binding energy; such spectra for the Fermi arc would provide more information than only plotting color scale images.
- 4) In Fig.5d, the authors plot an energy dispersion map as evidence for the Weyl cones. However, the data are hidden by orange dotted lines, so I wonder if the current data validates the Weyl cones. The authors need a better presentation to justify their claim. I also request that the authors plot the calculational dispersion side by side with this dispersion data to compare and discuss these.
- 5) The concept of engineering the Weyl state with Fe films on W is excellent. However, the authors require a more detailed discussion of how the wave function of W orbital is hybridized or mixed to that of Fe in the energy-momentum space to generate the band structure with the Weyl state, not only showing the band calculation results.
- 6) In Fig. S2, the authors present the surface state of 12MLs (comparable to bulk Fe), which could be topologically nontrivial according to the previous theoretical studies. It, however, is not well discussed whether or not this state is consistent with the previous theoretical suggestion of Weyl states in the bulk Fe. In addition, I request that the authors discuss the relationship between the bulk topological state and the Fermi arc of 2D Weyl state: how the former surface state turns to the latter when eventually influenced by the substrate of W.
- 7) Please add information on energy and momentum resolutions both for the spin-resolved and spin-integrated measurements in the method section.

If the above issues are appropriately addressed and reflected in the revised manuscript, I will

consider the manuscript for publication in Nature communications.

Reviewer #2:

Remarks to the Author:

This paper provides very interesting results regarding the mixed Berry curvature in the 2D ferromagnet (2ML Fe/W). It is demonstrated by showing a coupled open-ended surface Fermi arcs connecting two Weyl points by using a spin- and momentum-resolved ARPES technique. I was also impressed by the result of the magnetization switching of the Fermi arcs demonstrated in Fig.3.

Nevertheless, I feel that there are some deficiencies in the manuscript to reach their conclusion. I would recommend it for publication in Nature Communications after the following points are properly answered.

- 1) Possible orbital symmetry of the surface states needs to be clarified. This experiment is performed by using polarized light. Then d orbitals with some specific symmetries are allowed to be observed in photoemission spectra due to Fermi's golden rule. As a result, some specific spin orientation is supposed to be detected. The correspondence between the calculated spin texture in Fig.4b and the experimental result in Fig.4a is not guaranteed unless this point is taken into account. The authors need to discuss which orbital couples to which spin as was extensively discussed in the literature for Bi₂Se₃ surface states [C. Jozwiak et al., Nature Phys 9, 293–298 (2013)/ Y. Cao et al., Nature Phys 9, 499–504 (2013)].
- 2) It is necessary to exclude a possibility of the final state effects on the circular dichroism in order to state. The CD can arise from the final state effect as is known for heavy element metals. How can the authors exclude this possibility, because the observed states can contain the states from the substrate?
- 3) The topological invariants such as Chern numbers are not clearly stated in this paper. The observed two Fermi arcs are reminiscent of those in TaAs surface [X. Y. Xu et al., Science 349, 613 (2015).] that represent a chiral charge of 2. I think it is worth noting the chiral charge as well and hopefully discussing the relationship with other Weyl semimetals.
- 4) In this experiment, the spin polarization is evaluated by choosing different pass energies of the electron analyzer but with much different spin sensitivities. How can this large difference of the spin sensitivity be overcome in order to correctly evaluate the spin polarizations? This can be described in the Supplementary information or in Method section of the main text.

Reviewer #3:

None

Manuscript Title: Spanning Fermi arcs in a two-dimensional magnet

In the revised manuscript all suggestions and comments by the referees has been considered. This has led to a couple of changes in the manuscript and supplementary information. Modified text is highlighted in blue colour.

Reply to referee #1

Referee #1 (Remarks to the Author):

*This manuscript presents the spin-polarized Fermi arc emerging in 2MLs Fe films on W(110) due to a magnetic Weyl state. This feature has been successfully demonstrated by ARPES with a momentum microscope method equipped with a 2D spin detector. Weyl states are first demonstrated by ARPES for TaAs with the inversion symmetry breaking. Currently, magnetic Weyl states are being intensively studied in the field of condensed matter physics. In the states, the time-reversal symmetry breaking is the trigger for the realization of Weyl states. The magnetic Weyl states have functionalities being controllable by external magnetic field thus more advantageous for applications over those of inversion symmetry breaking systems. Such magnetic Weyl states have been experimentally demonstrated for 3D crystals so far. **The current work is unique in two ways: one is that Weyl states are generated in the form of films with two-dimensionality. The second is that the sample is made of conventional magnetic elements Fe even though it is assisted by W substrate with strong spin-orbit interaction. The data has very high quality, and the Fermi arc featuring Weyl states is convincingly revealed. In particular, the switching of spin polarization by flipping the magnetization direction is an excellent demonstration of engineering topological state with conventional magnetic metals which was achieved owing to the state-of-arts ARPES technique.***

We sincerely thank the referee for careful reviewing of our manuscript and the positive evaluation of our work as “very high quality,” “convincingly revealed” and “an excellent demonstration.” The referee has correctly pointed out the core message of our study and the importance of our results. This is extremely important and encouraging.

I will request the following issues to be addressed before considering publication of the manuscript:

We thank the referee for the valuable remarks that contributed substantially to the clarity of our manuscript. In the revised manuscript, several additional explanations and clarifications have been introduced. We are confident that our manuscript has been considerably improved by these revisions and is now ready for publication in Nature Communications. A point-by-point response to all the comments, remarks and questions is provided below.

1) In the 2 MLs Fe films on W(110), not only time-reversal symmetry but inversion symmetry are broken. From the current manuscript, I could not figure out the influence of each symmetry breaking on the spin-polarized Fermi arc. The authors should discuss this in detail.

We appreciate the referee’s stimulating questions since it shows concretely that this work is distinct from earlier studies and has far-reaching consequences for the interpretation of the Fermi arcs at the surface

of a 2D magnet. One of our co-author's paper [13 of the manuscript, Fig. 6f] has investigated the most simple model system, in which a free-standing sheet with preserved inversion symmetry but broken time-reversal symmetry. In that case, indeed, the pairs of Weyl nodes and the formation of a one-dimensional Fermi arc are predicted to span along the magnetization axis, θ .

There is, however, an important difference of our study, compared to this simplified model system: since the Fe layers are grown on the W substrate, simultaneously, the inversion symmetry as well as the time-reversal symmetry are broken. As shown in the Fig. 5, if symmetry is low enough, a pair of the Weyl nodes can distribute in the k_x - k_y plane, where the gap closing near the Fermi level emerges at $\theta \sim 88^\circ$. In that case, also the open loop Fermi arcs span in the k_x - k_y plane, as clearly observed in our experimental results.

In the revised manuscript the influence of time-reversal and inversion symmetries is now clarified as follows:

“When inversion symmetry would be preserved, ...” [page 10]

2) Throughout the entire manuscript, the arguments are given only with the presentation of photoelectron intensity maps. I request that the authors also plot up and down spin-resolved spectra along momentum cuts across the Fermi arc and discuss the details on the k_y component in Fig. 3 and k_x component in Fig.4.

We agree with the referee that line profiles of the spin resolved intensities across the Fermi arcs will help to get a better quantitative picture of the intricate polarization changes. Indeed such spectra can be readily extracted from the spin-resolved 2D momentum maps. Following the referee's recommendation, we now provide the measured spin-up and spin-down spectra along momentum cuts across the upper, middle, and lower part of the Fermi arc on the P_y and P_x components, as shown in the new Supplementary Figure 6 of the revised manuscript.

As we have already discussed in the manuscript, in particular the P_x polarization component shows a pronounced variation along the arc in Fig. 4. This behaviour can be also observed in the spin resolved spectra in the new Supplementary Fig. 6. A particular pronounced variation of the spin-up and spin-down intensities can be observed for P_x , comparing the sections across the upper and lower part of the Fermi arc. In the left branch of the arc (see black arrows for the arc location) an opposite spin polarization is observed. In the middle section cutting along the $k_y=0$ symmetry line, the spin polarization vanishes, i.e., the arc has equal spin-up and spin-down intensity contributions. A variation in the right arc branch is less pronounced. This result indicates a significant non-collinearity of the spin texture along the arc and is consistent with our theoretical calculations shown in Fig. 4b.

In the revised manuscript the new Supplementary Figure 6 is included. In addition the discussion of the non-collinearity is extended as follows:

“The variation of the P_x and P_y spin components in the upper and lower part of the arc can be seen quantitatively in horizontal spin resolved intensity profiles across the arc (see Supplementary Fig. 6 and Supplementary Note 6). ...” [page 7]

Additionally, the new Supplementary Figure 6 and the Supplementary Note 6 have been added in the revised manuscript

3) I also wonder if the authors also measured the spin-polarized spectra as a function of binding energy; such spectra for the Fermi arc would provide more information than only plotting color scale images.

A measurement of spin resolved spectra as a function of binding energy or E vs. k maps, as suggested by the referee, would represent an extremely demanding experiment for the investigated complex ferromagnet due to the intricate non-collinear variation of the spin polarization as a function of the momentum vector. It is thus not immediately clear which additional information can be gained from individual energy resolved spectra at single (k_x, k_y) points.

We nevertheless agree with the referee, that the energy dependence of the spin polarization is an interesting question. In the revised manuscript we therefore now also discuss the energy dependence of the Fermi arc spin polarization, and we have included a new Supplementary Figure 7 showing the calculated spin-resolved band structure along $\bar{H}-\bar{\Gamma}-\bar{H}$ on the P_y component with magnetization direction M_y .

The co-propagating surface arcs are shown. As discussed in our reply to question (2), above, the significant non-collinear spin texture along the arcs leads to a zero crossing of the P_y component in the right arc close to the $\bar{H}-\bar{\Gamma}-\bar{H}$ line. Consequently, P_y becomes zero in the calculated energy cut, being consistent with our calculations in Fig. 4b.

In particular, this result shows that the spin texture along the Fermi arcs is best described as a function of k_x and k_y in the spin polarization maps in Figs. 3b and 4, as well as the momentum resolved spectra in the new Supplementary Fig. 6. As a function of the energy, however, characteristic features like the zero crossing of the spin in the right Fermi arc, as discussed here, stay unchanged.

In the revised manuscript this is clarified in the Supplementary Note 6 in the paragraph starting from:

“The non-collinear spin texture of the Fermi arcs is found to lead to a pronounced variation of the spin direction in the k_x - k_y plane, while no significant variation with the binding energy was observed. ...” [page 3 of the supplementary information]

In addition the new Supplementary Figure 7 has been included in the supplementary information.

4) In Fig.5d, the authors plot an energy dispersion map as evidence for the Weyl cones. However, the data are hidden by orange dotted lines, so I wonder if the current data validates the Weyl cones. The authors need a better presentation to justify their claim. I also request that the authors plot the calculational dispersion side by side with this dispersion data to compare and discuss these.

We fully agree with the referee that the Weyl cone dispersion in the printed Fig. 5d is difficult to see. Followed by the referee's suggestions, we have now removed the orange dotted lines in one of the degenerate points in Fig. 5d. At the same time we have revised the presentation of the data in Fig. 5d to

show an enlarged part of the dispersion around the region of interest at the Fermi level. In addition, we have rearranged Fig. 5 to show the calculated dispersion of the Weyl cone side by side with the measured data, as shown in Fig. 5c and 5d. Moreover, we have strengthened the explicit description regarding the binding energy where the degeneracy points appear in calculated and experimental data.

In the revised manuscript the detailed binding energies of the calculated (at E_F) and measured ($E_F - 200\text{meV}$) Weyl node locations are now explicitly given and discussed. In the revised manuscript this now reads as:

“In good agreement with the calculation, the appearance of the degeneracy points is confirmed by our experimental data. Figure 5d shows the measured spectral function, where the degeneracy points appear at a slightly lower binding energy of 200 meV below the Fermi level. This location is still close enough to the Fermi level ...” [page 10]

5) The concept of engineering the Weyl state with Fe films on W is excellent. However, the authors require a more detailed discussion of how the wave function of W orbital is hybridized or mixed to that of Fe in the energy-momentum space to generate the band structure with the Weyl state, not only showing the band calculation results.

In contrast to the 2D band structure of a free standing Fe film, the wave functions of 2ML Fe/W(110) are hybridized and extended into the continuum of the Bloch states of the W substrate. As demonstrated in the Supplementary Fig. 3, no Fermi arcs emerge for free-standing Fe(110) monolayers and the W(110) substrate. Their Fermi surface contours show closed loop surface states as expected in ordinary metals. These calculations provide direct evidences that the topological change in spin-polarized Fermi surface of Fe is mediated by the hybridization of the wave functions due to the SOC brought about by the heavy-metal substrate.

The changed topology of the Fe Fermi surface sheets can be understood as follows: due to the presence of magnetic exchange interaction, the spin degeneracy is lifted in ferromagnetic Fe. Without SOC, the majority (\uparrow) and minority (\downarrow) bands are decoupled and can not interact. The hybridization of the Fe wave functions with the heavy-metal substrate tungsten leads to a strong SOC contribution, such that majority and minority spin states can mix at the crossing points. As a result, the spin-orbit-induced local gap leads to a discontinuity of the formerly closed contours of the majority and minority Fermi surface sheets of Fe, contravening that the iso-energy contour at the Fermi level of any conventional metal needs to be closed by a well-defined quasiparticle dispersion. As a result Fermi arcs are spanned across the gap opening as outlined in Fig. 1d.

The hybridisation of different Fermi surface sheets due to the introduction of large SOC, thus, can be seen as the driving force for the topological transition in the Fermi surface. Despite the two-dimensional character of the magnetic layer, topological degeneracy points then arise in the mixed (k_x, k_y, θ) phase space (see Fig. 5 and Supplementary Note 5), such that the open parts of the Fermi surface get connected by the experimentally observed giant Fermi arcs.

In the revised manuscript, the discussion on how the Weyl states are created by virtue of hybridization with the tungsten substrate has been extended along the lines noted above, in the paragraph starting with:

“Figure 1d illustrates how SOC leads to the emergence of the Fermi arcs in the iron layers. Without sizeable SOC, majority and minority Fermi surface contours can intersect without hybridization (left panel). The hybridization of Fe wave functions with the heavy-metal substrate tungsten” [page 4]

6) In Fig. S2, the authors present the surface state of 12MLS (comparable to bulk Fe), which could be topologically nontrivial according to the previous theoretical studies. It, however, is not well discussed whether or not this state is consistent with the previous theoretical suggestion of Weyl states in the bulk Fe.

As predicted by theory, bcc Fe is a topological metal that has two non-trivial electron pockets enclosing a Weyl point [27 of the manuscript]. The measured electron pockets in Fig. S2 are consistent with the theoretical prediction of electron packets, which enclose a Weyl point, and consequently being topologically non-trivial.

In the revised manuscript this has been clarified by:

“In our experiments we observe these topological electron pockets in bulk Fe (see Supplementary Fig.2 and Supplementary Note 2), in good agreement with previous reports [26,27]” [page 4]

In addition, I request that the authors discuss the relationship between the bulk topological state and the Fermi arc of 2D Weyl state: how the former surface state turns to the latter when eventually influenced by the substrate of W.

As correctly recognized by the referee, the substrate plays a crucial role in promoting the non-trivial features of the hidden topology in Fe. As clearly shown in Figs. 1 and 2 of the manuscript, the topological change of the spin-polarized Fermi surface sheets is then mediated by hybridization of the wave functions with the heavy-metal substrate.

As discussed above, bcc Fe has two topologically non-trivial electron pockets. The (110) surface of bulk bcc Fe presents topological Fermi-arcs resonances attached to the non-trivial electron pockets [27 of the manuscript]. Due to its metallic character, the surface Brillouin zone is almost entirely filled by the projected bulk states. These surface states thus overlap with the projection of the bulk states present at the Fermi level. In addition, the relatively weak spin-orbit coupling in bulk Fe gives rise to unstable Fermi-arc surface resonances. Indeed, the overlap with the bulk states together with the weak SOC of Fe hinders the emergence of stable Fermi arc surface states in bulk Fe, which thus have not been observed in an experiment so far.

We note that the location of the Fermi arcs on Fe/W(110) corresponds to the region where the two non-trivial electron pockets reside in bulk Fe. It has been shown that around this region the inclusion of spin-orbit coupling generates band touchings that act as sources and sinks of Berry curvature [26 of the

manuscript]. For the case of Fe grown on W(110), the stronger spin-orbit coupling inherited from the substrate changes the connectivity of the Fermi surface that results in a larger separation of the disconnected Fermi surface sheets and consequently promotes the emergence of the giant and stable Fermi arcs in Fe/W(110).

In the revised manuscript the role of the non-trivial electron pockets in bulk Fe has been clarified by:

“The relatively weak SOC in bulk Fe and the overlap with bulk states gives rise to only unstable Fermi-arc surface resonances, which thus have not been observed in an experiment so far [27]. Due to the increased SOC ...” [page 4]

7) Please add information on energy and momentum resolutions both for the spin-resolved and spin-integrated measurements in the method section.

In our experiments the energy resolution for the spin-resolved and spin-integrated measurements is about 0.05 eV. The momentum resolution for spin-integrated measurements is $\sim 0.02 \text{ \AA}^{-1}$. Since the W(100) crystal is used to project a spin-filtered 2D image onto a position sensitive detector, for spin-resolved measurements, the momentum resolution is determined by properties of the W(100) crystal. For Figs. 1b,c and Fig.3, the momentum resolution is 0.05 \AA^{-1} . Due to a smaller 2D field of view, for Fig.4a the momentum resolution is 0.02 \AA^{-1} . More details about the momentum resolution for spin-resolved measurements can be found in reference [31] of the manuscript.

The respective resolution values have been clarified in the revised manuscript in the methods section, in the paragraph starting from:

“In all measurements the energy resolution of the momentum microscope was set to 50 meV. The momentum resolution ...” [page 12]

If the above issues are appropriately addressed and reflected in the revised manuscript, I will consider the manuscript for publication in Nature communications.

We thank the referee for pointing out above questions, which have helped us to improve the clarity of our presentation. After having followed all recommendations by the referee in detail we are confident that our revised manuscript is now ready for publication in Nature Communications.

Reply to referee #2

Referee #2 (Remarks to the Author):

This paper provides very interesting results regarding the mixed Berry curvature in the 2D ferromagnet (2ML Fe/W). It is demonstrated by showing a coupled open-ended surface Fermi arcs connecting two Weyl points by using a spin- and momentum-resolved ARPES technique. I was also impressed by the result of the magnetization switching of the Fermi arcs demonstrated in Fig.3.

We sincerely thank the referee for positive evaluation of our work a “very interesting” and mentioning that “I was also impressed by the result of the magnetization switching of the Fermi arcs...” Indeed, we believe that the “switching of the Fermi arc states” that is highlighted here by the referee might become an important platform for directly interfacing with topological properties by external parameters. Such a novel control mechanism is likely to stimulate the discovery of novel device concepts by switchable topology, and open new research directions. We are therefore delighted to see that the referee shares our excitement and recognises this notable property that we firstly describe here.

Nevertheless, I feel that there are some deficiencies in the manuscript to reach their conclusion. I would recommend it for publication in Nature Communications after the following points are properly answered.

In the revised manuscript, therefore several additional explanations and clarifications have been introduced. We are confident that our manuscript has been considerably improved by these revisions and is now ready for publication in Nature Communications. A point-by-point response to all the comments, remarks and questions is provided below.

1) Possible orbital symmetry of the surface states needs to be clarified. This experiment is performed by using polarized light. Then d orbitals with some specific symmetries are allowed to be observed in photoemission spectra due to Fermi's golden rule. As a result, some specific spin orientation is supposed to be detected. The correspondence between the calculated spin texture in Fig.4b and the experimental result in Fig.4a is not guaranteed unless this point is taken into account. The authors need to discuss which orbital couples to which spin as was extensively discussed in the literature for Bi₂Se₃ surface states [C. Jozwiak et al., Nature Phys 9, 293–298 (2013)/ Y. Cao et al., Nature Phys 9, 499–504 (2013)].

In 3d metallic ferromagnets, most of the states are nearly pure spin states, namely, either spin-up (majority) or spin-down (minority) states with respect to the sample magnetization. The strong spin-orbit coupling in the region of the Fermi arcs, mediated through hybridization with substrate states, promotes a significant non-collinearity of the spin texture along the arcs. As a consequence, a complicated spin configuration is observed as being distinct from the spin-momentum locked surface states in Bi₂Se₃.

For the Dirac surface states in Bi₂Se₃, the strong spin-orbit coupling explicitly mixes the in-plane p_x and p_y orbitals with the p_z orbitals, in which a superposition of orbital wave functions coupled with the corresponding spin textures [e.g., see Meyerheim and Tusche, Phys. Status Solidi RRL 12, 1800078 (2018), Ref. 46 of the revised manuscript]. This phenomena is not unique for topological insulators. Our previous study has also shown the pronounced p -orbital dependence of the spin-texture in the case of a Rashba semiconductor [Maaß et. al. Nature Communications, 7, 11621 (2016), Ref. 45 of the revised

manuscript]. It follows that s-polarized light mainly excites electrons from the p_y orbital, while the p_x and p_z orbitals are excited by p-polarized light. As a result, a strong dependence of the photoelectron spin polarization on the polarization of the incoming light can be shown.

Differing from above materials, for the case of Fe/W, the Fermi-arc states are mainly composed of d orbitals. Particularly, one branch of an arc is of a dominant of d_{xy} orbital character, and another one composes mainly with d_{yz} orbitals. Since the arc states is mainly dominated by one orbital, it would not be expected to show a pronounced optical selection-rule effects on the photoelectron spin polarization.

Most importantly, we have performed first-principles calculations in order to verify that the experimentally probed spin-polarization correctly reflects the ground-state spin-texture in the arc states. The measured spin-polarization maps in Figs. 3b and 4a show the respective P_y and P_x components. Being ground-state calculations, the theoretical results in Fig. 4b are independent from the respective choice of the polarization of incoming light, and show a remarkably good agreement with the measured polarization in the arcs. Therefore the applicability of our experimental results to probe the spin-texture of the Fermi arcs is verified by our first-principles calculations.

In the revised manuscript the orbital symmetries are clarified. The revised part now reads:

“Furthermore, in contrast to topological surface states on many topological insulators and related layered materials, where the spin texture is locked to different p-orbital symmetries [45,46], the Fermi-arc states of 2ML Fe/W(110) are composed of d orbitals. In particular, our first principles calculations show ...”
[starting on page 7]

Additionally, in the revised manuscript the new references [45] ad [46] have been added.

2) It is necessary to exclude a possibility of the final state effects on the circular dichroism in order to state. The CD can arise from the final state effect as is known for heavy element metals. How can the authors exclude this possibility, because the observed states can contain the states from the substrate?

The referee correctly points out that the circular dichroism (CD), in some cases, does not solely reflect the ground state of the sample. In detail, the measured photoemission intensity I_α is proportional to $|M_\alpha|^2$, where the matrix element $|M_\alpha|^2 = \langle \mathbf{p} | O_\alpha | \Psi(\mathbf{k}) \rangle$ describes the excitation of an electron from an initial state $|\Psi(\mathbf{k})\rangle$ to a final state $\langle \mathbf{p} |$. The dipole operator, O_α , consists of the vector potential of $\alpha=L$ (left) or R (right) circularly polarized photons. The CD signals then arises from the coupling between the initial and final states [e.g. see Schönhense, Phys. Scr. T31, 255-275 (1990)].

For understanding the CD maps it is important to note that the surface arcs show the largest CD value among all other states. The largest magnitude of CD reaches 100% for the surface arcs, and remains unchanged with various photon, and binding energies. As shown in Fig. 2d of the manuscript, the two co-propagating surface arcs are the sharpest observed features in the CD signal, while all other states are very much blurred with substantially lower CD magnitude. The final state $\langle \mathbf{p} |$ in the above expression of the matrix element, though, is located at energies of $\sim 45\text{eV}$ above the vacuum level ($h\nu=50\text{eV}$). The unoccupied states at this energy strongly experience broadening (e.g., see Ref. [33] of the manuscript and references therein). In general, a CD that mainly originated from continuum final states leads to a

strongly broadened CD spectral widths. The remarkably sharp and large CD signal of the surface arcs, thus, can be mainly attributed to the initial state of the photoemission process, and reflects a property of the Fermi arc states.

In particular, it has been recently shown that the CD of conventional Weyl semimetals is sensitive to the orbital angular momentum of the topological states and can reflect the chirality of the respective wave functions [Ünzelmann et al., Nat. Commun. 12, 3650 (2021)]. With the considerations above, our results suggest that the observed strong (up to 100%) CD provides a fingerprint of the topological surface arc states, also in the case of the mixed topology of 2ML Fe.

In the revised manuscript this the role of the CD has been clarified as outlined above. The revised text now reads:

“The two co-propagating surface arcs are the sharpest observed features in the CD signal in Fig. 2d in contrast to the other states with relatively broad CD spectral features and substantially lower CD magnitude. Since the final states, that in principle might contribute to the CD signal, are significantly broadened ...” [page 8]

and:

“In particular, it recently has been shown that the CD of Fermi arc states of conventional Weyl semimetals ...” [page 9]

Additionally we added the new reference Ref. 49 [Ünzelmann et al., Nat. Commun. 12, 3650 (2021)].

3) The topological invariants such as Chern numbers are not clearly stated in this paper. The observed two Fermi arcs are reminiscent of those in TaAs surface [X. Y. Xu et al., Science 349, 613 (2015).] that represent a chiral charge of 2. I think it is worth noting the chiral charge as well and hopefully discussing the relationship with other Weyl semimetals.

The referee correctly points out that for many conventional Weyl semimetals, one would expect an integer chiral charge to be associated with the corresponding Weyl points in the regular 3D momentum space. In the present case of a mixed topology, the topological charges are located in the mixed phase space (k_x, k_y, θ). The presence of these topological charges becomes immediately clear by looking at the Berry curvature field ($-\Omega_{yy}^{\hat{m}k}$, $\Omega_{yx}^{\hat{m}k}$, Ω_{xy}^{kk}) in Fig. 5b, which shows a monopole-like source of the curvature field.

In conventional Weyl semimetals it is generally believed that the number of Fermi arcs corresponds to the chiral charge C of the Weyl point to which the arcs are attached. As pointed out by the referee in [X. Y. Xu et al., Science 349, 613 (2015).] the authors apply this concept to TaAs and find $C=\pm 2$. For various Type-II Weyl semimetals like MoTe_2 , this leads to a $C=\pm 1$ and with Fermi arc [e.g., see Deng et. al, Nat. Phys. 12, 1105-1110 (2016), and Jiang et. al, Nat. Commun. 8, 13973 (2017)]. Applying this concept to the present case of the mixed Weyl points we thus would expect a charge $C=\pm 2$.

In the revised manuscript a discussion regarding the chiral charge of the mixed Weyl point and a comparison to conventional Weyl semimetals has been included, in the paragraph starting from:

For conventional 3D Weyl semimetals it is generally believed that the number of Fermi arcs corresponds to the chiral charge C of the Weyl point to which the arcs are attached. For instance, TaAs was found to host Weyl points with chiral charge ...” [page 10]

In the revised manuscript the new references Ref. 50 [X. Y. Xu et al., Science 349, 613 (2015)], Ref. 51 [Deng et. al, Nat. Phys. 12, 1105-1110 (2016)] and Ref. 52 [Jiang et. al, Nat. Commun. 8, 13973 (2017)] have been included.

4) In this experiment, the spin polarization is evaluated by choosing different pass energies of the electron analyzer but with much different spin sensitivities. How can this large difference of the spin sensitivity be overcome in order to correctly evaluate the spin polarizations? This can be described in the Supplementary information or in Method section of the main text.

The referee points out a possible misunderstanding of the way how spin resolved measurements are evaluated. In general, spin resolved photoemission intensities are always derived from two independent measurements. In a conventional, e.g., Mott or SP-LEED, spin detector this would be measurements from two different counters on the left and right side of the detector. One need to note that the spin sensitivity of these two independent measurement always needs to be different in order to derive a spin polarization. In this simple example, the sensitivities of a left and right detector would be assumed to be exactly reversed, e.g., $S_L = -S_R = S$. Then the polarization is deduced from a measured asymmetry by the

simple relation $P = \frac{A}{S}$.

In the case of an imaging spin filter [31 of the manuscript] two independent measurements are recorded on the same detector, whereas the respective spin sensitivity is changed by altering the electron impact energy on the W(100) crystal. Images were recorded after reflection at a scattering energy alternating between 26.5 eV and 30.5 eV, switching the spin sensitivity S of the detector between 42% and 5%, respectively. since now the spin sensitivity is not exactly reversed between these two measurements, a generalized expression for the observed asymmetry needs to be used that accounts for the two different spin sensitivities. From the measured images, the spin polarization at every (k_x, k_y) point in a momentum disc (e.g., see Figs. 1,3,4) is given by

$$P(k_x, k_y) = \frac{I_{26.5\text{eV}}(k_x, k_y) - I_{30.5\text{eV}}(k_x, k_y)}{S_{26.5\text{eV}} \cdot I_{30.5\text{eV}}(k_x, k_y) - S_{30.5\text{eV}} \cdot I_{26.5\text{eV}}(k_x, k_y)} ,$$

where I_{ES} denotes the measured intensity image at scattering energy ES normalized by the respective reflectivity as measured from the clean Cu(100) surface. Details on the analysis procedure and the derivation of this formula can be found in Ref. [32 of the manuscript]. From this equation it becomes clear that a meaningful spin polarization P of the photoelectrons can be derived as long as the two measurements were performed at scattering energies that have sufficiently different spin sensitivities.

We note that above equation represent a generalized expression of the asymmetry. For instance, for the case of the left and right detectors of a conventional Mott and SP-LEED spin detector, the spin sensitivity is reversed, corresponding to here $S_{26.5\text{eV}} = -S_{30.5\text{eV}} = S$. In this case, the above equation can be then

simplified to $P = \frac{A}{S}$, with the measured asymmetry A . This formula, however, neglects instrumental asymmetries that are present in real systems, where equal spin sensitivities can usually not be assumed [Cacho et al., Rev. Sci. Instrum. 80, 043904 (2009)]. In the case of the imaging spin filter, on the other hand, instrumental influences of S are already included in the equation, such that no further corrections are required.

In the revised manuscript the procedure of the analysis of the spin polarization has been outlined in the methods section, starting from:

“From these images, the spin polarization at every (k_x, k_y) point in a momentum map at a certain binding energy (e.g., Figs. 1 and 3-4 in the main manuscript) is derived as ...” [page 12]

After having followed all recommendations by all referee in detail we are confident that our revised manuscript is now ready for publication in Nature Communications.

Reviewers' Comments:

Reviewer #1:

Remarks to the Author:

In the previous round of review, I requested several issues to be more clearly described, which I found all to be properly addressed in the revised manuscript. From the previous round, my impression has been the same in that the results are impressive in both aspects that the Weyl states are nicely controlled by the external magnetic field, and those are convincingly demonstrated with state-of-art equipment. The revised manuscript showing those results with further clarification should have a large impact on the community, so I am now happy to accept this version for publication in Nature communications.

Reviewer #2:

Remarks to the Author:

The authors have revised their manuscript substantially in order to reflect what is raised by the two referees. I find that all the concerns raised by the two referees have been properly answered. I recommend the current manuscript be published in Nature Communication.